# Large-Margin Convex Polytope Machine

**Alex Kantchelian**    **Michael Carl Tschantz**    **Ling Huang**[†]
**Peter L. Bartlett**    **Anthony D. Joseph**    **J. D. Tygar**

UC Berkeley – {akant|mct|bartlett|adj|tygar}@cs.berkeley.edu
[†]Datavisor – ling.huang@datavisor.com

## Abstract

We present the Convex Polytope Machine (CPM), a novel non-linear learning algorithm for large-scale binary classification tasks. The CPM finds a large margin convex polytope separator which encloses one class. We develop a stochastic gradient descent based algorithm that is amenable to massive datasets, and augment it with a heuristic procedure to avoid sub-optimal local minima. Our experimental evaluations of the CPM on large-scale datasets from distinct domains (MNIST handwritten digit recognition, text topic, and web security) demonstrate that the CPM trains models faster, sometimes several orders of magnitude, than state-of-the-art similar approaches and kernel-SVM methods while achieving comparable or better classification performance. Our empirical results suggest that, unlike prior similar approaches, we do not need to control the number of sub-classifiers (sides of the polytope) to avoid overfitting.

## 1    Introduction

Many application domains of machine learning use massive data sets in dense medium-dimensional or sparse high-dimensional spaces. These domains also require near real-time responses in both the prediction and the model training phases. These applications often deal with inherent non-stationarity, thus the models need to be constantly updated in order to catch up with drift. Today, the de facto algorithm for binary classification tasks at these scales is linear SVM. Indeed, since Shalev-Shwartz *et al.* demonstrated both theoretically and experimentally that large margin linear classifiers can be efficiently trained at scale using stochastic gradient descent (SGD), the Pegasos [1] algorithm has become a standard building tool for the machine learning practitioner.

We propose a novel algorithm for Convex Polytope Machine (CPM) separation exhibiting superior empirical performance to existing algorithms, with running times on a large dataset that are up to five orders of magnitude faster. We conjecture that worst case bounds are independent of the number $K$ of faces of the convex polytope and state a theorem of loose upper bounds in terms of $\sqrt{K}$.

In theory, as the VC dimension of $d$-dimensional linear separators is $d + 1$, a linear classifier in very high dimension $d$ is expected to have a considerable expressiveness power. This argument is often understood as "everything is separable in high dimensional spaces; hence linear separation is good enough". However, in practice, deployed systems rarely use a single naked linear separator. One explanation for this gap between theory and practice is that while the probability of a single hyperplane perfectly separating both classes in very high dimensions is high, the resulting classifier margin might be very small. Since the classifier margin also accounts for the generalization power, we might experience poor future classification performance in this scenario.

Figure 1a provides a two-dimensional example of a data set that has a small margin when using a single separator (solid line) despite being linearly separable and intuitively easily classified. The intuition that the data is easily classified comes from the data naturally separating into three clusters

with two of them in the positive class. Such clusters can form due to the positive instances being generated by a collection of different processes.

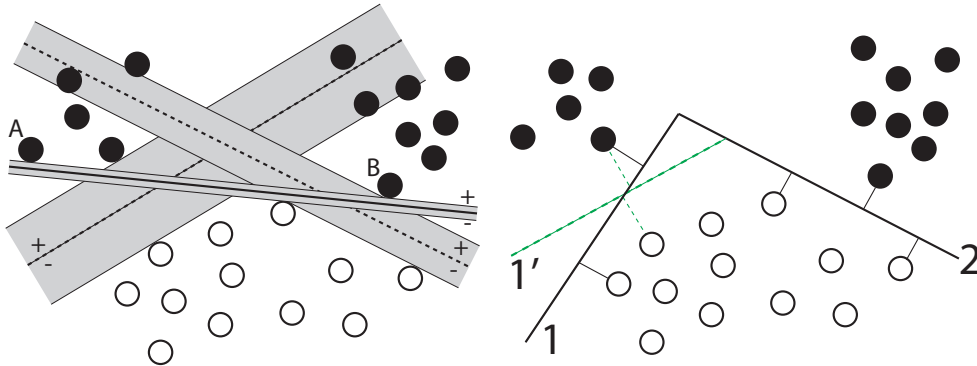

(a) Instances are perfectly linearly separable (solid line), although with small margin due to positive instances (A & B) having conflicting patterns. We can obtain higher margin by separately training two linear sub-classifiers (dashed lines) on left and right clusters of positive instances, each against all the negative instances, yielding a prediction value of the maximum of the sub-classifiers.

(b) The worst-case margin is insensitive to wiggling of sub-classifiers having non-minimal margin. Sub-classifier 2 has the smallest margin, and sub-classifier 1 is allowed to freely move without affecting $\delta^{\text{WC}}$. For comparison, the largest-margin solution $1'$ is shown (dashed lines).

Figure 1: Positive (●) and negative (○) instances in continuous two dimensional feature space.

As Figure 1a shows, a way of increasing the margins is to introduce two linear separators (dashed lines), one for each positive cluster. We take advantage of this intuition to design a novel machine learning algorithm that will provide larger margins than a single linear classifier while still enjoying much of the computational effectiveness of a simple linear separator. Our algorithm learns a bounded number of linear classifiers simultaneously. The global classifier will aggregate all the sub-classifiers decisions by taking the maximum sub-classifier score. The maximum aggregation has the effect of assigning a positive point to a unique sub-classifier. The model class we have intuitively described above corresponds to convex polytope separators.

In Section 2, we present related work in convex polytope classifiers and in Section 3, we define the CPM optimization problem and derive loose upper bounds. In Section 4, we discuss a Stochastic Gradient Descent-based algorithm for the CPM and perform a comparative evaluation in Section 5.

## 2 Related Work

Fischer focuses on finding the optimal polygon in terms of the number of misclassified points drawn independently from an unknown distribution using an algorithm with a running time of more than $O(n^{12})$ where $n$ is the number of sample points [2]. We instead focus on finding good, not optimal, polygons that generalize well in practice despite having fast running times. Our focus on generalization leads us to maximize the margin, unlike this work, which actually minimizes it to make their proofs easier. Takacs proposes algorithms for training convex polytope classifiers based on the smooth approximation of the maximum function [3]. While his algorithms use smooth approximation during training, it uses the original formula during prediction, which introduces a gap that could deteriorate the accuracy. The proposed algorithms achieve similar classification accuracy to several nonlinear classifiers, including KNN, decision tree and kernel SVM. However, the training time of the algorithms is often much longer than those nonlinear classifiers (e.g., an order of magnitude longer than ID3 algorithm and eight times longer than kernel SVM on CHESS DATASET), diminishing the motivation to use the proposed algorithms in realistic setting. Zhang *et al.* propose an Adaptive Multi-hyperplane Machine (AMM) algorithm that is fast during both training and prediction, and capable of handling nonlinear classification problems [4]. They develop an iterative algorithm based on the SGD method to search for the number of hyperplanes and train the model. Their experiments on several large data sets show that AMM is nearly as fast as the state-of-the-art linear SVM solver, and achieves classification accuracy somewhere between linear and kernel

SVMs. Manwani and Sastry propose two methods for learning polytope classifiers, one based on logistic function [5], and another based on perceptron method [6], and propose alternating optimization algorithms to train the classifiers. However, they only evaluate the proposed methods with a few small datasets (with no more than 1000 samples in each), and do not compare them to other widely used (nonlinear) classifiers (e.g., KNN, decision tree, SVM). It is unclear how applicable these algorithms are to large-scale data. Our work makes three significant contributions over their work, including 1) deriving the formulation from a large-margin argument and obtaining a regularization term which is missing in [6], 2) safely restricting the choice of assignments to only positive instances, leading to a training time optimization heuristic and 3) demonstrating higher performance on non-synthetic, large scale datasets, when using two CPMs together.

## 3   Large-Margin Convex Polytopes

In this section, we derive and discuss several alternative optimization problems for finding a large-margin convex polytope which separates binary labeled points of $\mathbb{R}^d$.

### 3.1   Problem Setup and Model Space

Let $\mathcal{D} = \{(\mathbf{x}^i, y^i)\}_{1 \leq i \leq n}$ be a binary labeled dataset of $n$ instances, where $\mathbf{x} \in \mathbb{R}^d$ and $y \in \{-1, 1\}$. For the sake of notational brevity, we assume that the $\mathbf{x}^i$ include a constant unitary component corresponding to a bias term. Our prediction problem is to find a classifier $c : \mathbb{R}^d \to \{-1, 1\}$ such that $c(\mathbf{x}^i)$ is a good estimator of $y^i$. To do so, we consider classifiers constructed from convex $K$-faced polytope separators for a fixed positive integer $K$. Let $\mathcal{P}_K$ be the model space of convex $K$-faced polytope separators:

$$\mathcal{P}_K = \left\{ f : \mathbb{R}^d \to \mathbb{R} \,\middle|\, f(\mathbf{x}) = \max_{1 \leq k \leq K} (\mathbf{W}\mathbf{x})_k, \mathbf{W} \in \mathbb{R}^{K \times d} \right\}$$

For each such function $f$ in $\mathcal{P}_K$, we can get a classifier $c_f$ such that $c_f(\mathbf{x})$ is 1 if $f(\mathbf{x}) > 0$ and $-1$ otherwise. This model space corresponds to a shallow single hidden layer neural network with a max aggregator. Note that when $K = 1$, $\mathcal{P}_1$ is simply the space of all linear classifiers. Importantly, when $K \geq 2$, elements of $\mathcal{P}_K$ are not guaranteed to have additive inverses in $\mathcal{P}_K$. As a consequence, the labels $y = -1$ and $y = +1$ are not interchangeable. Geometrically, the negative class remains enclosed within the convex polytope while the positive class lives outside of it, hence the label asymmetry.

To construct a classifier without label asymmetry, we can use two polytopes, one with the negative instances on the inside the polytope to get a classification function $f_-$ and one with the positive instances on the inside to get $f_+$. From these two polytopes, we construct the classifier $c_{f_-, f_+}$ where $c_{f_-, f_+}(\mathbf{x})$ is 1 if $f_-(\mathbf{x}) - f_+(\mathbf{x}) > 0$ and $-1$ otherwise.

To better understand the nature of the faces of a single polytope, for a given polytope $\mathbf{W}$ and a data point $\mathbf{x}$, we denote by $z_{\mathbf{W}}(\mathbf{x})$ the index of the maximum sub-classifier for $\mathbf{x}$:

$$z_{\mathbf{W}}(\mathbf{x}) = \underset{1 \leq k \leq K}{\operatorname{argmax}} (\mathbf{W}\mathbf{x})_k$$

We call $z_{\mathbf{W}}(\mathbf{x})$ the *assigned sub-classifier* for instance $\mathbf{x}$. When clear from context, we drop $\mathbf{W}$ from $z_{\mathbf{W}}$. We also use the notation $\mathbf{W}_k$ to designate the $k$-th row of $\mathbf{W}$, which corresponds to the $k$-th face of the polytope, or the $k$-th sub-classifier. Hence, $\mathbf{W}_{z(\mathbf{x})}$ identifies the separator assigned to $\mathbf{x}$.

We now pursue a geometric large-margin based approach for formulating the concrete optimization problem. To simplify the notations and without loss of generality, we suppose that $\mathbf{W}$ is row-normalized such that $||\mathbf{W}_k|| = 1$ for all $k$. We also initially suppose our dataset is perfectly separable by a $K$-faced convex polytope.

### 3.2   Margins for Convex Polytopes

When $K = 1$, the problem reduces to finding a good linear classifier and only a single natural margin $\delta$ of the separator exists [7]:

$$\delta_{\mathbf{W}} = \min_{1 \leq i \leq n} y^i \mathbf{W}_1 \mathbf{x}^i$$

Maximizing $\delta_{\mathbf{W}}$ yields the well known (linear) Support Vector Machine. However, multiple notions of margin for a $K$-faced convex polytope with $K \geq 2$ exist. We consider two.

Let the *worst case margin* $\delta_{\mathbf{W}}^{\text{WC}}$ be the smallest margin of any point to the polytope. Over all the $K$ sub-classifiers, we find the one with the minimal margin to the closest point assigned to it:

$$\delta_{\mathbf{W}}^{\text{WC}} = \min_{1 \leq i \leq n} y^i \mathbf{W}_{z(\mathbf{x}^i)} \mathbf{x}^i = \min_{1 \leq k \leq K} \min_{i:z(\mathbf{x}^i)=k} y^i \mathbf{W}_k \mathbf{x}^i$$

The worst case margin is very similar to the linear classifier margin but suffers from an important drawback. Maximizing $\delta^{\text{WC}}$ leaves $K-1$ sub-classifiers wiggling while over-focusing on the sub-classifier with the smallest margin. See Figure 1b for a geometrical intuition.

Thus, we instead focus on the *total margin*, which measures each sub-classifier's margin with respect to just its assigned points. The total margin $\delta_{\mathbf{W}}^{\text{T}}$ is the sum of the $K$ sub-classifiers margins:

$$\delta_{\mathbf{W}}^{\text{T}} = \sum_{k=1}^{K} \min_{i:z(\mathbf{x}^i)=k} y^i \mathbf{W}_k \mathbf{x}^i$$

The total margin gives the same importance to the $K$ sub-classifier margins.

### 3.3 Maximizing the Margin

We now turn to the question of maximizing the margin. Here, we provide an overview of a smoothed but non-convex optimization problem for maximizing the total margin. The appendix provides a step-by-step derivation.

We would like to optimize the margin by solving the optimization problem

$$\max_{\mathbf{W}} \delta_{\mathbf{W}}^{\text{T}} \quad \text{subject to} \quad \|\mathbf{W}_1\| = \cdots = \|\mathbf{W}_K\| = 1 \tag{1}$$

Introducing one additional variable $\zeta_k$ per classifier, problem (1) is equivalent to:

$$\max_{\mathbf{W},\zeta} \sum_{k=1}^{K} \zeta_k \quad \text{subject to} \quad \forall i, \zeta_{z(\mathbf{x}^i)} \leq y^i \mathbf{W}_{z(\mathbf{x}^i)} \mathbf{x}^i \tag{2}$$
$$\zeta_1 > 0, \ldots, \zeta_K > 0$$
$$\|\mathbf{W}_1\| = \cdots = \|\mathbf{W}_K\| = 1$$

Considering the unnormalized rows $\mathbf{W}_k/\zeta_k$, we obtain the following equivalent formulation:

$$\max_{\mathbf{W}} \sum_{k=1}^{K} \frac{1}{\|\mathbf{W}_k\|} \quad \text{subject to} \quad \forall i, 1 \leq y^i \mathbf{W}_{z(\mathbf{x}^i)} \mathbf{x}^i \tag{3}$$

When $y = -1$ and $z(\mathbf{x}^i)$ satisfy the margin constraint in (3), we have that the constraint holds for every sub-classifier $k$ since $y^i \mathbf{W}_k \mathbf{x}^i$ is minimal at $k = z(\mathbf{x}^i)$. Thus, when $y = -1$, we can enforce the constraint for all $k$. We can also smooth the objective into a convex, defined everywhere one by minimizing the sum of the inverse squares of the terms instead of maximizing the sum of the terms. We obtain the following smoothed problem:

$$\min_{\mathbf{W}} \sum_{k=1}^{K} \|\mathbf{W}_k\|^2 \quad \text{subject to} \quad \forall i : y^i = -1, \forall k \in \{1, \ldots, K\}, 1 + \mathbf{W}_k \mathbf{x}^i \leq 0 \tag{4}$$
$$\forall i : y^i = +1, 1 - \mathbf{W}_{z(\mathbf{x}^i)} \mathbf{x}^i \leq 0 \tag{5}$$

The objective of the above program is now the familiar $L_2$ regularization term $\|\mathbf{W}\|^2$. The negative instances constraints (4) are convex (linear functions), but the positive terms (5) result in non-convex constraints because of the instance-dependent assignment $z$. As for the Support Vector Machine, we can introduce $n$ slack variables $\xi_i$ and a regularization factor $C > 0$ for the common case of noisy, non-separable data. Hence, the practical problem becomes:

$$\min_{\mathbf{W},\xi} \|\mathbf{W}\|^2 + C \sum_{i=1}^{n} \xi_i \quad \text{subject to} \quad \forall i : y^i = -1, \forall k \in \{1, \ldots, K\}, 1 + \mathbf{W}_k \mathbf{x}^i \leq \xi_i \geq 0 \tag{6}$$
$$\forall i : y^i = +1, 1 - \mathbf{W}_{z(\mathbf{x}^i)} \mathbf{x}^i \leq \xi_i \geq 0$$

Following the same steps, we obtain the following problem for maximizing the worst-case margin. The only difference is the regularization term in the objective function which becomes $\max_k \|\mathbf{W}_k\|^2$ instead of $\|\mathbf{W}\|^2$.

**Discussion.** The goal of our relaxation is to demonstrate that our solution involves two intuitive steps, including (1) assigning positive instances to sub-classifiers, and (2) solving a collection of SVM-like sub-problems. While our solution taken as a whole remains non-convex, this decomposition isolates the non-convexity to a single intuitive assignment problem that is similar to clustering. This isolation enables us to use intuitive heuristics or clustering-like algorithms to handle the non-convexity. Indeed, in our final form of Eq. (6), if the optimal assignment function $z(\mathbf{x}^i)$ of positive instances to sub-classifiers were known and fixed, the problem would be reduced to a collection of perfectly independent convex minimization problems. Each such sub-problem corresponds to a classical SVM defined on all negative instances and the subset of positive instances assigned by $z(\mathbf{x}^i)$. It is in this sense that our approach optimizes the total margin.

### 3.4 Choice of $K$, Generalization Bound for CPM

Assuming we can efficiently solve this optimization problem, we would need to adjust the number $K$ of faces and the degree $C$ of regulation. The following result gives a preliminary generalization bound for the CPM. For $B_1, \ldots, B_k \geq 0$, let $\mathcal{F}_{K,B}$ be the following subset of the set $\mathcal{P}_K$ of convex polytope separators:

$$\mathcal{F}_{K,B} = \left\{ f : \mathbb{R}^d \to \mathbb{R} \,\middle|\, f(\mathbf{x}) = \max_{1 \leq k \leq K} (\mathbf{W}\mathbf{x})_k, \mathbf{W} \in \mathbb{R}^{K \times d}, \forall k, \|\mathbf{W}_k\| \leq B_k \right\}$$

**Theorem 1.** *There exists some constant $A > 0$ such that for all distributions $P$ over $\mathcal{X} \times \{-1, 1\}$, $K$ in $\{1, 2, 3, \ldots\}$, $B_1, \ldots, B_k \geq 0$, and $\delta > 0$, with probability at least $1 - \delta$ over the training set $(\mathbf{x}_1, y_1), \ldots, (\mathbf{x}_n, y_n) \sim P$, any $f$ in $\mathcal{F}_{K,B}$ is such that:*

$$P(yf(\mathbf{x}) \leq 0) \leq \frac{1}{n} \sum_{i=1}^{n} \max(0, 1 - y_i f(\mathbf{x}_i)) + A \frac{\sum_k B_k}{\sqrt{n}} + \sqrt{\frac{\ln(2/\delta)}{2n}}$$

This is a uniform bound on the 0-1 risk of classifiers in $\mathcal{F}_{K,B}$. It shows that with high probability, the risk is bounded by the empirical hinge loss plus a capacity term that decreases in $n^{-1/2}$ and is proportional to the sum of the sub-classifier norms. Note that as we have $\sum_k \|\mathbf{W}_k\| \leq \sqrt{K}\|\mathbf{W}\|$, the capacity term is essentially equivalent to $\sqrt{K}\|\mathbf{W}\|$. As a comparison, the generalization error has been previously shown to be proportional to $K\|\mathbf{W}\|$ in [4, Thm. 2]. In practice, this bound is very loose as it does not explain the observed absence of over fitting as $K$ gets large. We experimentally demonstrate this phenomenon in Section 5. We conjecture that there exists a bound that must be independent of $K$ altogether. The proof of Theorem 1 relies on a result due to Bartlett *et al.* on Rademacher complexities. We first prove that the Rademacher complexity of $F_{K,B}$ is in $O(\sum_k B_k/\sqrt{n})$. We then invoke Theorem 7 of [8] to show our result. The appendix contains the full proof.

## 4 SGD-based Learning

In this section, we present a Stochastic Gradient Descent (SGD) based learning algorithm for approximately solving the total margin maximization problem (6). The choice of SGD is motivated by two factors. First, we would like our learning technique to efficiently scale to several million instances of sparse high dimensional space. The sample-iterative nature of SGD makes it a very suitable candidate to this end [9]. Second, the optimization problem we are solving is non-convex. Hence, there are potentially many local optima which might not result in an acceptable solution. SGD has recently been shown to work well for such learning problems [10] where we might not be interested in a global optimum but only a good enough local optimum from the point of view of the learning problem.

Problem (6) can be expressed as an unconstrained minimization problem as follow:

$$\min_{\mathbf{W}} \sum_{i:y^i=-1} \sum_{k=1}^{K} [1 + \mathbf{W}_k \mathbf{x}^i]_+ + \sum_{i:y^i=+1} [1 - \mathbf{W}_{z(\mathbf{x}^i)} \mathbf{x}^i]_+ + \lambda \|\mathbf{W}\|^2$$

where $[x]_+ = \max(0, x)$ and $\lambda > 0$. This form reveals the strong similarity with optimizing $K$ unconstrained linear SVMs [1]. The difference is that although each sub-classifier is trained on

all the negative instances, positive instances are associated to a unique sub-classifier. From the unconstrained form, we can derive the stochastic gradient descent Algorithm 1. For the positive instances, we isolate the task of finding the assigned sub-classifier $z$ to a separate procedure ASSIGN. We use the Pegasos inverse schedule $\eta_t = 1/(\lambda t)$.

Because the optimization problem (6) is non-convex, a pure SGD approach could get stuck in a local optimum. We found that pure SGD gets stuck in low-quality local optima in practice. These optima are characterized by assigning most of the positive instances to a small number of sub-classifiers. In this configuration, the remaining sub-classifiers serve no purpose. Intuitively, the algorithm clustered the data into large "super-clusters" ignoring the more subtle sub-clusters comprising the larger super-clusters. The large clusters represent an appealing local optima since breaking one down into sub-clusters often requires transitioning through a patch of lower accuracy as the sub-classifiers realign themselves to the new cluster boundaries. We may view the local optima as the algorithm underfitting the data by using too simple of a model. In this case, the algorithm needs encouragement to explore more complex clusterings.

**Algorithm 1** Stochastic gradient descent algorithm for solving problem (6).

> **function** SGDTRAIN($\mathcal{D}, \lambda, T, (\eta_t), h$)
>   Initialize $\mathbf{W} \in \mathbb{R}^{K \times d}$, $\mathbf{W} \leftarrow \mathbf{0}$
>   **for** $t \leftarrow 1, \ldots, T$ **do**
>     Pick $(\mathbf{x}, y) \in \mathcal{D}$
>     **if** $y = -1$ **then**
>       **for** $k \leftarrow 1, \ldots, K$ **do**
>         **if** $\mathbf{W}_k \mathbf{x} > -1$ **then**
>           $\mathbf{W}_k \leftarrow \mathbf{W}_k - \eta_t \mathbf{x}$
>     **else if** $y = +1$ **then**
>       $z \leftarrow \text{argmax}_k \mathbf{W}_k \mathbf{x}$
>       **if** $\mathbf{W}_z \mathbf{x} < 1$ **then**
>         $z \leftarrow \text{ASSIGN}(\mathbf{W}, \mathbf{x}, h)$
>       $\mathbf{W}_z \leftarrow \mathbf{W}_z + \eta_t \mathbf{x}$
>
>     $\mathbf{W} \leftarrow (1 - \eta_t \lambda) \mathbf{W}$
>   **return** $\mathbf{W}$

With this intuition in mind, we add a term encouraging the algorithm to explore higher entropy configurations of the sub-classifiers. To do so, we use the entropy of the random variable $Z = \text{argmax}_k \mathbf{W}_k \mathbf{x}$ where $\mathbf{x} \sim \mathcal{D}^+$, a distribution defined on the set of all positive instances as follows. Let $n_k$ be the number of positive instances assigned to sub-classifier k, and $n$ be the total number of positive instances. We define $\mathcal{D}^+$ as the empirical distribution on $\left(\frac{n_1}{n}, \frac{n_2}{n}, \ldots, \frac{n_k}{n}\right)$. The entropy is zero when the same classifier fires for all positive instances, and maximal at $\log_2 K$ when every classifier fires on a $K^{-1}$ fraction of the positive instances. Thus, maximizing the entropy encourages the algorithm to break down large clusters into smaller clusters of near equal size.

We use this notion of entropy in our heuristic procedure for assignment, described in Algorithm 2. ASSIGN takes a predefined minimum entropy level $h \geq 0$ and compensates for disparities in how positive instances are assigned to sub-classifiers, where the disparity is measured by entropy. When the entropy is above $h$, there is no need to change the natural $\text{argmax}_k \mathbf{W}_k \mathbf{x}$ assignment. Conversely, if the current entropy is below $h$, then we pick an assignment which is guaranteed to increase the entropy. Thus, when $h = 0$, there is no adjustment made. It keeps a dictionary UNADJ mapping the previous points it has encountered to the unadjusted assignment that the natural argmax assignment would had made at the time of encountering the point. We write UNADJ $+ (\mathbf{x}, k)$ to denote the new dictionary $U$ such that $U[\mathbf{v}]$ is equal to $k$ if $\mathbf{v} = \mathbf{x}$ and to UNADJ$[\mathbf{v}]$ otherwise. Dictionary UNADJ keeps track of the assigned positives per sub-classifiers, and serves to estimate the current entropy in the configuration without needing to recompute every prior point's assignment.

## 5   Evaluation

We use four data sets to evaluate the CPM: (1) an MNIST dataset consisting of labeled handwritten digits encoded in $28 \times 28$ gray scale pictures [11, 12] (60,000 training and 10,000 testing instances); (2) an MNIST8m dataset consisting of 8,100,000 pictures obtained by applying various random deformations to MNIST training instances MNIST [13]; (3) a URL dataset [12] used for malicious URL detection [14] (1.1 million training and 1.1 million testing instances in a very large dimensional space of more than 2.3 million features); and (4) the RCV1-bin dataset [12] corresponding to a binary classification task (separating *corporate* and *economics* categories from *government* and *markets* categories [15]) defined over the RCV1 dataset of news articles (20,242 training and 677,399 testing instances). Since our main focus is on binary classification, for the two MNIST datasets we evaluate

distinguishing 2's from any other digit, which we call MNIST-2 and MNIST8m-2. With thirty times more testing than training data, the RCV1-bin dataset is a good benchmark for over fitting issues.

## 5.1 Parameter Tuning

All four datasets have well defined training and testing subsets and to tune each algorithms meta-parameters ($\lambda$ and $h$ for the CPM, $C$ and $\gamma$ for RBF-SVM, and $\lambda$ for AMM), we randomly select a fixed validation subset from the training set (10,000 instances for MNIST-2/MNIST8m-2; 1,000 instances for RCV1-bin/URL).

---

**Algorithm 2** Heuristic maximum assignment algorithm. The input is the current weight matrix $\mathbf{W}$, positive instance $\mathbf{x}$, and the desired assignment entropy $h \geq 0$.

---
Initialize UNADJ$\leftarrow \{\}$
**function** ASSIGN$(\mathbf{W}, \mathbf{x}, h)$
    $k_{\mathsf{unadj}} \leftarrow \operatorname{argmax}_k \mathbf{W}_k \mathbf{x}$
    **if** ENTROPY$($UNADJ$+ (\mathbf{x}, k_{\mathsf{unadj}})) \geq h$ **then**
        $k_{\mathsf{adj}} \leftarrow k_{\mathsf{unadj}}$
    **else**
        $h_{\mathsf{cur}} \leftarrow$ ENTROPY$($UNADJ$)$
        $\mathbf{K}_{\mathsf{inc}} \leftarrow \{k\colon$ENTROPY$($UNADJ$+(\mathbf{x}, k)) > h_{\mathsf{cur}}\}$
        $k_{\mathsf{adj}} \leftarrow \underset{k \in \mathbf{K}_{\mathsf{inc}}}{\operatorname{argmax}} \mathbf{W}_k \mathbf{x}$
    UNADJ $\leftarrow$ UNADJ $+ (\mathbf{x}, k_{\mathsf{unadj}})$
    **return** $k_{\mathsf{adj}}$

---

For the CPM, we use a double-sided CPM as described in section 3.1, where both CPMs share the same meta-parameters. We start by fixing a number of iterations $T$ and a number of hyperplanes $K$ which will result in a reasonable execution time, effectively treating these parameters as a computational budget, and we experimentally demonstrate that increasing either $K$ or $T$ always results in a decrease of the testing error. Once these are selected, we let $h = 0$ and select the best $\lambda$ in $\{T^{-1}, 10 \times T^{-1}, \ldots, 10^4 \times T^{-1}\}$. We then choose $h$ from $\{0, \log K/10, \log 2K/10, \ldots, \log 9K/10\}$, effectively performing a one-round coordinate descent on $\lambda, h$. To test the effectiveness of our empirical entropy-driven assignment procedure, we mute the mechanism by also testing with $h = 0$.

The AMM has three parameters to adjust (excluding $T$ and the equivalent of $K$), two of which control the weight pruning mechanism and are left set at default values. We only adjust $\lambda$. Contrary to the CPM, we do not observe AMM testing error to strictly decrease with the number of iterations $T$. We observe erratic behavior and thus we manually select the smallest $T$ for which the mean validation error appears to reach a minimum. For RBF-SVM, we use the LibSVM [16] implementation and perform the usual grid search on the parameter space.

## 5.2 Performance

Unless stated otherwise, we used one core of an Intel Xeon E5 (3.2Ghz, 64GB RAM) for experiments. Table 1 presents the results of experiments and shows that the CPM achieves comparable, and at times better, classification accuracy than the RBF-SVM, while working at a relatively small and constant computational budget. For the CPM, $T$ was up to 32 million and $K$ ranged from 10 to 100. For AMM, $T$ ranged from 500,000 to 36 million. Across methods, the worst execution time is for the MNIST8m-2 task, where a 512 core parallel implementation of RBF-SVM runs in 2 days [17], and our sequential single-core algorithm runs in less than 5 minutes. The AMM has significantly larger errors and/or execution times. For small training sets such as MNIST-2 and RCV1-bin, we were not able to achieve consistent results, regardless of how we set $T$ and $\lambda$, and we conjecture that this is a consequence of the weight pruning mechanism. The results show that our empirical entropy-driven assignment procedure for the CPM leads to better solutions for all tasks. In the RCV1-bin and MNIST-2 tasks, the improvement in accuracy from using a tuned entropy parameter is 31% and 21%, respectively, which is statistically significant.

We use the MNIST8m-2 task to the study the effects of tuning $T$ and $K$ on the CPM. We first choose a grid of values for $T, K$ and for a fixed regularization factor $C$ and $h = 0$, we train a model for each point of the parameter grid, and evaluate its performance on the testing set. Note that for $C$ to remain constant, we adjust $\lambda = \frac{1}{CT}$. We run each experiment 5 times and only report the mean accuracy. Figure 2 shows how this mean error rate evolves as a function of both $T$ and $K$. We observe two phenomena. First, for any value $K > 1$, the error rate decreases with $T$. Second, for large enough values of $T$, the error rate decreases when $K$ increases. These two experimental

| | MNIST-2 | | MNIST8m-2 | | URL | | RCV1-bin | |
|---|---|---|---|---|---|---|---|---|
| | Error | Time | Error | Time | Error | Time | Error | Time |
| CPM | $0.38 \pm 0.028$ | 2m | $0.30 \pm 0.023$ | 4m | $1.32 \pm 0.012$ | 3m | $2.82 \pm 0.059$ | 2m |
| CPM $h=0$ | $0.46 \pm 0.026$ | 2m | $0.35 \pm 0.034$ | 4m | $1.35 \pm 0.029$ | 3m | $3.69 \pm 0.156$ | 2m |
| RBF-SVM | 0.35 | 7m | $0.43^*$ | $2d^{**}$ | Timed out in 2 weeks | | 3.7 | 46m |
| AMM | $2.83 \pm 1.090$ | 1m | $0.38 \pm 0.024$ | 1hr | $2.20 \pm 0.067$ | 5m | $15.40 \pm 6.420$ | 1m |

\* for unadjusted parameters [17]        \*\* running on 512 processors [17]

Table 1: Error rates and running times (include both training and testing periods) for binary tasks. Means and standard deviations for 5 runs with random shuffling of the training set.

observations validate our treatment of both $K$ and $T$ as budgeting parameters. The observation about $K$ also provides empirical evidence of our conjecture that large values of $K$ do not lead to overfitting.

### 5.3 Multi-class Classification

We performed a preliminary multi-class classification experiment using the MNIST/MNIST8m datasets. There are several approaches for building a multi-class classifier from a binary classifier [18, 19, 20]. We used a one-vs-one approach where we train $\binom{10}{2} = 45$ one-vs-one classifiers and classify by a majority vote rule with random tie breaking. While this approach is not optimal, it provides an approximation of achievable performance. For MNIST, comparing CPM to RBF-SVM, we achieve a testing error of $1.61 \pm 0.019$ and for the CPM and of $1.47$ for RBF-SVM, with running times of 7m20s and 6m43s, respectively. On MNIST8m we achieve an error of $1.03 \pm 0.074$ for CPM (2h3m) and of $0.67$ (8 days) for RBF-SVM as reported by [13].

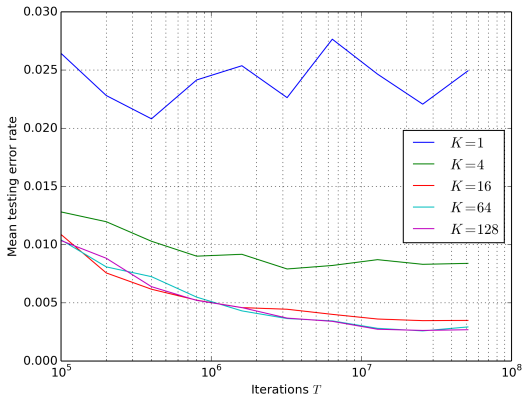

Figure 2: Error rate on MNIST8m-2 as a function of $K, T$. $C = 0.01$ and $h = 0$ are fixed.

## 6 Conclusion

We propose a novel algorithm for Convex Polytope Machine (CPM) separation that provides larger margins than a single linear classifier, while still enjoying the computational effectiveness of a simple linear separator. Our algorithm learns a bounded number of linear classifiers simultaneously. On large datasets, the CPM outperforms RBF-SVM and AMM, both in terms of running times and error rates. Furthermore, by not pruning the number of sub-classifiers used, CPM is algorithmically simpler than AMM. CPM avoids such complications by having little tendency to overfit the data as the number $K$ of sub-classifiers increases, shown empirically in Section 5.2.

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
