[Supplementary Material]

# Appendix for Large-Margin Convex Polytope Machine

**Alex Kantchelian    Michael Carl Tschantz    Ling Huang[†]**
**Peter L. Bartlett    Anthony D. Joseph    J. D. Tygar**

UC Berkeley – {akant|mct|bartlett|adj|tygar}@cs.berkeley.edu
[†]Datavisor – ling.huang@datavisor.com

## 1   Details of Maximizing the Margin

We now turn to the question of maximizing the margin. We show the step-by-step derivation a smoothed but non-convex optimization problem for maximizing the total margin.

$$\max_{\mathbf{W}} \delta_{\mathbf{W}}^{\mathrm{T}} \tag{1}$$
$$\|\mathbf{W}_1\| = \cdots = \|\mathbf{W}_K\| = 1$$

Introducing one additional variable $\zeta_k$ per classifier, problem (1) is equivalent to:

$$\max_{\mathbf{W},\zeta} \sum_{k=1}^{K} \zeta_k \tag{2}$$
$$\forall i, \zeta_{z(\mathbf{x}^i)} \le y^i \mathbf{W}_{z(\mathbf{x}^i)} \mathbf{x}^i$$
$$\zeta_1 > 0, \ldots, \zeta_K > 0$$
$$\|\mathbf{W}_1\| = \cdots = \|\mathbf{W}_K\| = 1$$

Considering the unnormalized rows $\mathbf{W}_k/\zeta_k$, we obtain the following equivalent formulation:

$$\max_{\mathbf{W}} \sum_{k=1}^{K} \frac{1}{\|\mathbf{W}_k\|} \tag{3}$$
$$\forall i, 1 \le y^i \mathbf{W}_{z(\mathbf{x}^i)} \mathbf{x}^i \tag{4}$$

When $y = -1$, $z(\mathbf{x}^i)$ satisfying the margin constraint (4) implies that the constraint holds for every sub-classifier $k$ since $y^i \mathbf{W}_k \mathbf{x}^i$ is minimal at $k = z(\mathbf{x}^i)$. Thus, when $y = -1$, we can enforce the constraint for all $k$ yielding the following equivalent problem:

$$\max_{\mathbf{W}} \sum_{k=1}^{K} \frac{1}{\|\mathbf{W}_k\|} \tag{5}$$
$$\forall i : y^i = -1, \forall k \in \{1, \ldots, K\}, 1 + \mathbf{W}_k \mathbf{x}^i \le 0$$
$$\forall i : y^i = +1, 1 - \mathbf{W}_{z(\mathbf{x}^i)} \mathbf{x}^i \le 0$$

Finally, we can relax the objective into a convex one by minimizing the sum of the inverse squares of the terms instead of maximizing the sum of the terms. We obtain the following smoothed problem:

$$\min_{\mathbf{W}} \sum_{k=1}^{K} \|\mathbf{W}_k\|^2 \tag{6}$$

$$\forall i : y^i = -1, \forall k \in \{1, \ldots, K\}, 1 + \mathbf{W}_k \mathbf{x}^i \leq 0 \tag{7}$$

$$\forall i : y^i = +1, 1 - \mathbf{W}_{z(\mathbf{x}^i)} \mathbf{x}^i \leq 0 \tag{8}$$

The objective (6) is now the familiar convex $L_2$ regularization term $\|\mathbf{W}\|^2$. The negative samples constraints (7) are convex (linear functions), but the positive terms (8) result in non-convex constraints because of the instance-dependent assignment $z$. As for the Support Vector Machine, we can introduce $n$ slack variables $\xi_i$ and a regularization factor $C > 0$ for the common case of noisy, non-separable data. Hence, the practical problem becomes:

$$\min_{\mathbf{W}, \xi} \|\mathbf{W}\|^2 + C \sum_{i=1}^{n} \xi_i \tag{9}$$

$$\forall i : y^i = -1, \forall k \in \{1, \ldots, K\}, 1 + \mathbf{W}_k \mathbf{x}^i \leq \xi_i$$

$$\forall i : y^i = +1, 1 - \mathbf{W}_{z(\mathbf{x}^i)} \mathbf{x}^i \leq \xi_i$$

$$\forall i, \xi_i \geq 0$$

Following the same steps, we obtain the following problem for maximizing the worst-case margin. The only difference is the regularization term in the objective function which becomes $\max_k \|\mathbf{W}_k\|^2$ instead of $\|\mathbf{W}\|^2$.

## 2   Proof of Theorem 1

The Rademacher complexity of $F_{K,B}$ is defined as

$$R_n(F_{K,B}) = \mathbb{E}_{\mathbf{x}} \mathbb{E}_{\epsilon} \left[ \sup_{f \in F_{K,B}} \left| \frac{1}{n} \sum_i \epsilon_i f(\mathbf{x}_i) \right| \right]$$

Where the $\epsilon_i$ are $\pm 1$ i.i.d. Bernoulli with probability 1/2. It is also possible to define the Gaussian Rademacher complexity of $F_{K,B}$ is as:

$$G_n(F_{K,B}) = \mathbb{E}_{\mathbf{x}} \mathbb{E}_{g} \left[ \sup_{f \in F_{K,B}} \left| \frac{1}{n} \sum_i g_i f(\mathbf{x}_i) \right| \right]$$

where the $g_i$s are i.i.d. standard normal variables.

By Lemma 4 in [1], there exists an absolute constant $c$ such that for every $F_{K,B}$ and $n$ we have $R_n(F_{K,B}) \leq cG_n(F_{K,B})$. Thus, we can provide a bound on the Gaussian Rademacher complexity. In our case, this can be directly done by invoking Theorem 14 of [1]. Indeed, $a_1, \ldots, a_k \mapsto \max(a_1, \ldots, a_k)$ is a Lipchitz with constant 1, thus $F_{K,B}$ can be viewed as the composition of the max function with the real valued classes of linear separators $F_i$ that are such that

$$F_i = \{\mathbf{x} \mapsto \langle \mathbf{W}, \mathbf{x} \rangle | \|\mathbf{W}\| \leq B_i\}$$

So we have that $G_n(F_{K,B}) \leq 2 \sum_{k=1}^{K} G_n(F_k)$. The Gaussian Rademacher complexities of each of these $F_k$s is bounded by $B_k/\sqrt{n}$ by a standard argument as follows:

$$G_n(F_k) = \mathbb{E}_{\mathbf{x}}\mathbb{E}_g \left[ \sup_{\|\mathbf{W}\| \leq B_k} \left| \frac{1}{n} \sum_i g_i \langle \mathbf{W}, \mathbf{x}_i \rangle \right| \right]$$

$$= \mathbb{E}_{\mathbf{x}}\mathbb{E}_g \left[ \sup_{\|\mathbf{W}\| \leq B_k} \frac{1}{n} \langle \mathbf{W}, \sum_{i=1}^n \mathbf{x}_i g_i \rangle \right]$$

$$= \mathbb{E}_{\mathbf{x}}\mathbb{E}_g \frac{B_k}{n} \| \sum_{i=1}^n \mathbf{x}_i g_i \|$$

$$\leq \mathbb{E}_{\mathbf{x}} \frac{B_k}{n} \sqrt{\mathbb{E}_g \| \sum_{i=1}^n \mathbf{x}_i g_i \|^2}$$

$$= \mathbb{E}_{\mathbf{x}} \frac{B_k}{n} \sqrt{\sum_{i=1}^n \| \mathbf{x}_i \|^2}$$

$$\leq \frac{B_k}{\sqrt{n}}$$

Hence, there exists a universal constant $A > 0$ such that

$$R_n(F_{K,B}) \leq A \sum_k G_n(F_k) = A \frac{\sum_k B_k}{\sqrt{n}}$$

Finally, we apply Theorem 7 [1] where $\phi$ is taken to be the hinge loss, and obtain the desired result.

## References

[1] P. L. Bartlett and S. Mendelson. Rademacher and Gaussian complexities: Risk bounds and structural results. *J. Mach. Learn. Res.*, 3:463–482, Mar. 2003.