[Reviews · NeurIPS 2014]

Submitted by Assigned_Reviewer_21

This paper presents a large-margin learning algorithm for convex polytope machines for binary classification tasks. The optimization objective is obtained from relaxing the problem of maximizing the total margin. A preliminary generalization bound regarding the objective function is presented. The paper proposes to use SGD to solve the optimization problem Eq. (6). The experiments, which are conducted on standard benchmarks, demonstrate convincing performance over baselines including RBF SVM.

It seems that the proposed formulation of optimization objective could be further clarified or improved. The paper claims to optimize the total margin (line 170-176). Indeed, the first objectives (Eq. (1) and Eq. (2)) directly maximize the total margin. These objective functions are non-convex, as pointed out in the paper. Therefore, the authors propose Eq. (3-5) and Eq. (6) as the objective function to be optimized, suggesting that they are the relaxation of Eq. (1) and Eq. (2). However, the quantitative connection between Eq. (3-6) and Eq. (1-2) is unclear to me and is unfortunately not clarified in the paper. It would be better if the paper develops or explains in more quantitative way how minimizing Eq. (3-5) will maximize the total margin. Furthermore, even after relaxation, the objective Eq. (6) is still non-convex. Thus I doubt whether the proposed relaxation has served its purpose well. In sum, the proposed objective function Eq. (6) seem to be a choice that is not well-justified, since (1) it is unclear how well it maximizes the total margin; (2) the relaxed objective is still non-convex.

The main theoretical result (Theorem 1) in the paper is proven using standard arguments of proving generalization bounds. The proof does not seem to contain much new techniques or insights; it would be better if the paper points out the novelties of its proof, if any.

The proposed SGD algorithm is heuristic and lacks theoretical underpinnings. The paper suggests that pure SGD does not work (due to the non-convexity) and therefore the authors add an entropy term to encourage smaller clusters. However, besides some intuitions, there are no formal analyses on how this entropy term will influence the objective function. Further, the paper does not describe clearly how the entropy is computed; line 278 says that it is computed by assuming x~D^+, but I cannot find the definition of _distribution_ D^+ in the text. While these details are missing or unclear, the paper includes some implementation details (which are very standard and well-known techniques) that add little value to the paper, e.g., maintaining scaling factors, using dense/sparse matrices, the internal data structure and using C++11. I think that the space may be better utilized if it is used to present more algorithm/model related details. Nevertheless, I appreciate the authors’ promise of releasing the source code to the community.

The experiment results shown in the paper are good. But many experimental results are omitted (including the influence of K on the rest of data sets) due to space constraints. These results (especially the influence of K) may be useful of understanding the advantages of large-margin convex polytope machines over ordinary SVMs. Perhaps the space of supplementary material can be utilized to present more experiments results.

The writing of this paper could be improved in a few places.
- line 248. “sparse high dimensional space”;
- line 265-266. “a pure SGD approach could get stuck in a local optimum. We found that pure SGD gets stuck in low-quality local optima in practice”;
Summary: This paper studies convex polytope machines for binary classification problems and proposes a large-margin learning algorithm based on the SGD algorithm assisted with some heuristics. A preliminary generalization bound is presented and the experiment results are encouraging. However, the choice of objective function is not well justified, and the learning algorithm lacks theoretical underpinnings. The main theorem in the paper seems to be proven using standard methods. Some details are missing from the paper.

Submitted by Assigned_Reviewer_23

In this paper, the authors propose the convex polytope machine (CPM) for binary classification. The idea (which is not new) is to use more than one hyperplane for the decision function. They derive a non-convex objective and use SGD for parameter estimation. They also give a generalization bound based on Rademacher complexities.
The proposed method performs better than a similar method called AMM.

One issue with the paper is that the authors use on 4 datasets in their experiment section. To be really convincing, the authors should use many more datasets than that.

Section 3.2: it was not clear to me whether "worst-case" and "total" margin are new margin definitions or not. Also, for comparison, indicate which margin definition does AMM use.

For reproducibility of the results, please indicate the values selected by
cross-validation for the number of classifiers K, the regularization parameter
lambda and the number of iterations T.

Did you use SGD with parameter averaging?

You should cite "Multiclass Classification with Multi-Prototype Support
Vector Machines", by Aiolli and Sperduti. This is the paper on which AMM is
based and deserves a citation.
Summary: A nice paper although the experiment section is a bit short.

Submitted by Assigned_Reviewer_42

--- Summary of the paper
This paper formulates binary classification as a convex polytope learning problem with the large margin principle, and
1. shows preliminary results on error bound,
2. proposes an efficient learning algorithm based on SGD with maximum entropy heuristics, and
3. empirically evaluates its performance and dependency on the number of faces K of the polytope to support the theoretical result.

-- Quality
I followed the mathematics of the paper, and they seem technically sound. The empirical results are nice and the proposed method can be one of the candidates for the classification method for large datasets.

-- Clarity
Basically easy to follow, but there are some sentences that I cannot make sense of.
Also, more importantly, the description of the algorithms are somewhat unclear and redundant.

-- Originality and significance
I'm not sure whether this formulation of the binary classification problem with large margin polytope separator is really novel because there are vast amount of related works and extensions since the invention of the SVM.
Assuming the proposed formulation is novel, I feel it is reasonable and the related learning theory is legitimate.
The obtained error bound is, though the author conjectures it can be more more tight, tighter than the one for the conventional problem (ANN).
The experimental results are promising and support the theoretical result.
However, the algorithm description impairs the quality of the paper (e.g., although ASSIGN is defined, argmax_k W_k x operation is explicitly used in several parts). Particularly, I didn't fully understand how the algorithm HEURISTICMAXASSIGN uses the list UNADJ.
Summary: Overall, the motivation is clear, methodologies are sound, and the results are reasonable and encouraging. With more clear description of methodology, I could score higher.
Author Feedback
Author rebuttal: We thank the reviewers for their attentive reading and insightful comments. We will amend the paper as follows: (1) clarify the optimization algorithm, especially for the entropy-based assignment function; (2) further explain on the derivation of the CPM formulation; (3) extend the experimental section with results on several more experiments (we have already worked with different binary problems derived from MNIST8m, a9a, and webspam, and are considering KDD2010) and report the meta-parameter values used; (4) compress the implementation details; and (5) cite Aiolli and Sperduti’s original paper.

Meanwhile, we provide our reviewers the following detailed information to clarify our work.

[Reviewer 21]
- Our general approach trades convexity for representability; thus we do not aim for a convex optimization problem. Instead, the goal of our relaxation is to reduce the problem to two intuitive steps:
(1) an assignment of positive instances to sub-classifiers, and
(2) a collection of SVM-like sub-classifiers.
While the problem taken as a whole remains non-convex, this decomposition isolates the non-convexity to a single intuitive assignment problem that is similar to clustering. This clean decomposition allows the employment of intuitive heuristics or clustering-like algorithms to handle the non-convexity.
- To see this, looking at in the final form of Eq. (6) or unconstrained form used in section 4, if the optimal assignment function z of positive instances to sub-classifiers were known and fixed, the problem would be reduced to a collection of perfectly independent convex minimization problems. Each such sub-problem corresponds to a classical SVM defined on all negative instances and the subset of positive instances assigned by z. It is in this sense that our approach optimizes the total margin.
- The key smoothing occurs in between Eq. (2) and Eq. (3), where instead of maximizing a sum of inverses, we minimize the sum of the squares (minimization of a convex objective to be easier to work with). If K=1, this is sound and leads to the SVM. The remaining of the paper shows both experimental and theoretical evidence in favor of this for K>1. This step takes the problem from one with difficult to reason about non-convexity to one where the non-contexity is isolated and intuitive.
- The entropy is computed over the empirical distribution (n_1/n,…,n_K/n) where n_k is the number of positive instances assigned to sub-classifier k and n is the total number of positive instances. (The notation D^+ refers to the empirical distribution of positive instances.) The idea is to better spread positive instances on the sub-classifiers (as measured by the above entropy), hopefully finding a better local minimum. We acknowledge that we can only demonstrate strong empirical evidence from diverse datasets in favor of this heuristic at this point, and are working on deriving theoretical justification for it.

[Reviewer 32]
- The definition of "total" margin is new, insofar it is specialized to the context of convex polytope separators. The traditional notion of classifier margin w.r.t. a dataset would in this case coincide with the "worst-case" margin, which suffers from the discussed drawback. Because of the multi-class formulation, the AMM does not have a direct geometric interpretation in terms of margin. Thus, it is hard to precisely answer the reviewer’s question about AMM. We however note that the AMM regularization term is the same as ours, namely ||W||^2, which suggests a total-margin-like behavior.
- We do not use parameter averaging, but agree it is an interesting idea to explore. To the best of our knowledge, theoretical and experimental results only exist for convex problems. Using it in our non-convex context warrants further investigation. Further, setting the time when averaging kicks-in might be non-trivial.

[Reviewer 42]
- To the best of our knowledge, the closest formulation to ours appears in a paper from
Manwani and Sastri: “Polyceptron: A Polyhedral Learning Algorithm” (citation [6]). We make four significant improvements and contributions over their work: (1) we show how to derive the formulation from a large-margin argument; (2) as a result, we obtain a regularization term which is missing in [6]; (3) we (safely) restrict the choice of assignments to only positive instances; (4) we demonstrate higher performance on non-synthetic, large scale datasets, when using two CPMs together.
- The HEURISTICMAXASSIGN function compensates for disparities in how positive instances are assigned to sub-classifiers. We use entropy as the quantitative measure of disparity. Thus, if the entropy is high enough, there is no need to change the natural max() assignment. Conversely, if the current entropy is below the required level, then we pick an assignment which is guaranteed to increase the entropy. In the paper, UNADJ keeps track of the assigned positives per sub-classifiers in order to compute the entropy at every stage.